# Selective enhancement of low-gamma activity by tACS improves phonemic processing and reading accuracy in dyslexia

Silvia Marchesotti[1]☉*, Johanna Nicolle[1]☉, Isabelle Merlet[2], Luc H. Arnal[1,3], John P. Donoghue[4], Anne-Lise Giraud[1]*

**1** Department of Neuroscience, University of Geneva, Geneva, Switzerland, **2** Univ Rennes, Inserm, LTSI, UMR 1099, Rennes, France, **3** Institut de l'Audition, Institut Pasteur, INSERM, Paris, France, **4** Brown University, Providence, Rhode Island, United States of America

☉ These authors contributed equally to this work.
* anne-lise.giraud@unige.ch (ALG); silvia.marchesotti@unige.ch (SM)

## Abstract

The phonological deficit in dyslexia is associated with altered low-gamma oscillatory function in left auditory cortex, but a causal relationship between oscillatory function and phonemic processing has never been established. After confirming a deficit at 30 Hz with electroencephalography (EEG), we applied 20 minutes of transcranial alternating current stimulation (tACS) to transiently restore this activity in adults with dyslexia. The intervention significantly improved phonological processing and reading accuracy as measured immediately after tACS. The effect occurred selectively for a 30-Hz stimulation in the dyslexia group. Importantly, we observed that the focal intervention over the left auditory cortex also decreased 30-Hz activity in the right superior temporal cortex, resulting in reinstating a left dominance for the oscillatory response. These findings establish a causal role of neural oscillations in phonological processing and offer solid neurophysiological grounds for a potential correction of low-gamma anomalies and for alleviating the phonological deficit in dyslexia.

**Data Availability Statement:** Data reported in this paper are publicly available at https://osf.io/6j49q/.

**Funding:** This study has been supported by the Wyss Center for Bio and Neuro Engineering (WCP-

Dyslexia is a frequent disorder of reading acquisition affecting up to 7% of schoolchildren and characterized by persistent difficulties with written material throughout adulthood. Identifying the neural bases of dyslexia to devise efficient treatments has motivated intense research in the last decades [1–3]. These treatments include behavioral auditory and reading training [4,5] and, more recently, noninvasive electrical brain stimulation [6–8], even though the exact underlying action mechanisms remain uncertain. From a neuroscience viewpoint, dyslexia poses an interesting challenge because it selectively affects one aspect of language, the mapping of phonemes onto graphemes, while leaving other cognitive domains intact, such as speech perception and production or visual and auditory processing [9]. Although several possible causes have been proposed for dyslexia [10], the predominant one is a phonological deficit, i.e., a difficulty in processing the sounds of language. The deficit primarily affects phonological

006, to A-LG, www.wysscenter.ch) and Swiss National Science Foundation (320030B_182855, to A-LG, www.snf.ch). The funders had no role in study design, data collection and analysis, decision to publish, or preparation of the manuscript.

**Competing interests:** The authors have declared that no competing interests exist.

**Abbreviations:** AM, amplitude-modulated; ASSR, auditory steady-state response; ECLA, Évaluation de Compétences de Lecture chez l'Adulte de plus de 16 ans (reading skills assessment for adults over age 16); EEG, electroencephalography; FDR, False Discovery Rate; ICA, independent component analysis; LTP, long-term potentiation; NMDA, N-methyl-D-aspartate; RAN, rapid automatized naming; ROI, region of interest; STG, superior temporal gyrus; STP, short-term plasticity; S/N, signal-to-noise ratio; tACS, transcranial alternating current stimulation; tDCS, transcranial direct current stimulation.

awareness, conscious access, representation and internal monitoring of speech sounds [11–13], the capacity to form rich categorical phonemic representations [13,14], and naming and verbal memory [15].

Unlike oral language, which arises through mere exposure, reading requires explicit learning, through which children become aware that the syllables they are used to parsing (e.g., via nursery rhymes) are made of smaller units, the phonemes. Learning to read consists of mapping these new basic phonological building blocks to specific visual symbols. This so-called phoneme–grapheme mapping is only possible if the child is able to match the sound associated with the visual symbol with his/her own phonemic inventory, made of infrasyllabic elements that can be taken out and replaced by another articulable sound [16]. At the acoustic level, critical phonemic contrasts (e.g., /t/ versus /d/ or /d/ versus /b/, etc.) are underpinned by rapid events (noise bursts, formant transitions, voicing, etc.), and grasping them requires auditory sampling at a rate allowing for their neural encoding as individual patterns. The optimal sampling rate should hence enable the integration of basic 30- to 40-ms acoustic segments, comprising roughly vocal tract occlusion and periocclusion cues such as voicing [17].

Theoretical models propose that neural oscillations in the 25–35 Hz range could be the basic speech sampling rate, from which would derive (in an evolutionary sense) the inferior bound of the phonemic temporal format, i.e., the shortest possible linguistic unit (approximately 25–30 ms) that can be individually represented [18]. Basic neurophysiological studies confirm that 2 sounds must be separated by at least 25 ms to be perceptually and neurally individualized [19]. In line with these observations, neuroimaging studies have repeatedly linked dyslexia with a deficit in oscillatory activity in the low-gamma band [20–24]. This deficit could be related to difficulties in the processing of rise time in amplitude-modulated (AM) sounds, typically associated with anomalies in slower neural oscillation ranges [24–26]. Young adults with dyslexia show a disrupted low-gamma 30-Hz response in left auditory cortex and abnormally strong responses at higher frequencies (around 40 Hz), suggesting that auditory sampling could be faster than in typical readers [22,27]. This double anomaly has been associated with a deficit in phonological processing (because of atypical phonemic format) and working memory (because of more basic phonemic elements per language unit to hold in memory [22]), respectively. Furthermore, consistent with atypical morphological asymmetry of temporal cortex in dyslexia [28], left dominance of the low-gamma auditory response is also disturbed. In both cases (anatomical and functional), the asymmetry anomaly is statistically related with the phonological deficit [22].

These findings converge toward a possible oscillation theory of phonemic construction and of the phonological deficit in dyslexia, originating primarily in atypical functioning of the left auditory cortex. This theory is worthy of investigation because it could offer an easy entry point for therapeutic interventions through noninvasive neuromodulation aiming at normalizing oscillatory function in auditory cortex [29]. Building on the hypothesis that disrupted low-gamma activity in left auditory cortex could be causally related to the phonological deficit in dyslexia, we tested whether restoring low-gamma oscillatory function in individuals with dyslexia could improve phonemic perception and, indirectly, reading performance.

To address this question, we used high-definition transcranial alternating current stimulation (tACS) with a 4 × 1 ring electrode configuration that was focal, painless, and frequency-specific under the assumption that stimulation at 30 Hz should boost neural activity at the same frequency [30,31]. To characterize the effect, the design included several other conditions (a sham stimulation, a 60-Hz tACS condition, a control group, and a delayed measure 1 hour after tACS in which the effects were expected to have largely disappeared). We conducted this neuromodulation study in a single-blind way in 30 adults (15 with dyslexia, 15 fluent readers, mean age 26.4 years, SD ± 8.1, range 18–47), involving for each subject 22 hours of

experimental procedures spread over 4 experimental days (S1 Fig). After a thorough assessment of dyslexia in all participants on day 1, they performed on days 2–4 a custom-designed battery of language tests (S1 Table) probing reading efficiency (speed and accuracy) and phonological processing (via pseudoword test, i.e., nonlexical word repetition, and spoonerism, i.e., inverting the first phoneme of a word pair). Prior to the language tests, we recorded auditory steady-state response (ASSR) to pure tones modulated in amplitude with a fixed frequency (range: 28–62 Hz) through a 64-channel electroencephalography (EEG) system. EEG data were also acquired prior and after tACS intervention in order to confirm reduced 30-Hz responses in individuals with dyslexia relative to readers without dyslexia and monitor the neurophysiological changes subsequently induced by tACS.

ASSRs and behavioral tests were performed before, immediately after, and 1 hour after 20 minutes of focal tACS ($\leq$2 mA stimulation) over left auditory cortex, a stimulation duration that is argued to be effective while minimizing potential side effects [32]. The 1 hour after tACS was included to both evaluate and control for potential long-term effect of the stimulation. The specificity of the 30-Hz intervention was further controlled using 2 within-subject conditions: an active control stimulation at 60 Hz probing whether the expected effects could be merely due to delivering alternating current and a placebo stimulation (sham condition) in which no current was delivered. Given the selective deficit in left auditory cortex in subjects with dyslexia, we expect focal tACS to partly restore the 30-Hz neural response in that specific region in the dyslexia group, inducing a benefit for phonemic processing, whereas such an effect is not expected either in controls or after 60-Hz stimulation in either group.

## Results

Before examining the effect of tACS on phonemic processing and reading, we first tested whether we could reproduce the 30-Hz deficit in the left auditory cortex that was previously shown in subjects with dyslexia [20,22,27]. This confirmation was a requirement for addressing whether tACS could effectively normalize neural activity. Hence, we analyzed the EEG auditory responses to AM sounds at frequencies between 28 and 62 Hz as measured before tACS. Auditory stimuli evoked typical waveforms with N100 and P200 auditory components [33] and topographies, with the strongest peak-to-peak response amplitude at FCz (Fig 1A). In a first approach, we restricted the EEG analysis to this electrode. In line with a large body of literature [34,35], the time–frequency transformation of the EEG signal during the steady-state interval (ASSR) peaked for auditory stimuli delivered with a 40-Hz amplitude modulation (Fig 2A). Because the 30-Hz response deficit in dyslexia was expected to be lateralized and to dominate in left auditory cortex [22], we further reconstructed the activity of neural generators responsible for the signal recorded on the scalp. Based on previous results [28,36], we performed a source space analysis on 2 regions of interest (ROIs) in each cerebral hemisphere, corresponding to auditory cortex and superior temporal gyrus (STG). Source reconstruction obtained from scalp recording also identified the strongest responses at these locations (S2 Fig). Although the group × hemisphere interaction was not significant ($F_{1,27} = 2.05$, $p = 0.16$, $\eta^2_p = 0.07$), we could confirm a significant deficit in 30-Hz activity in the left auditory cortex in subjects with dyslexia relative to controls, ($T_{27} = 2.1$, $p < 0.05$, d = 0.77; Fig 1C, left), a prerequisite to a local intervention aiming at boosting this activity. We also confirmed that the effect was not present in the right auditory cortex ($T_{27} = 0.068$, $p > 0.05$, d = 0.02; Fig 1C, right), indicating that the deficit was lateralized. Further, there was no 30-Hz deficit in the superior temporal region surrounding auditory cortex in either hemisphere, confirming previous observations [22].

Having established that subjects with dyslexia had a selective deficit in neural oscillations at 30 Hz, we examined the impact of tACS on behavior. To directly compare changes in

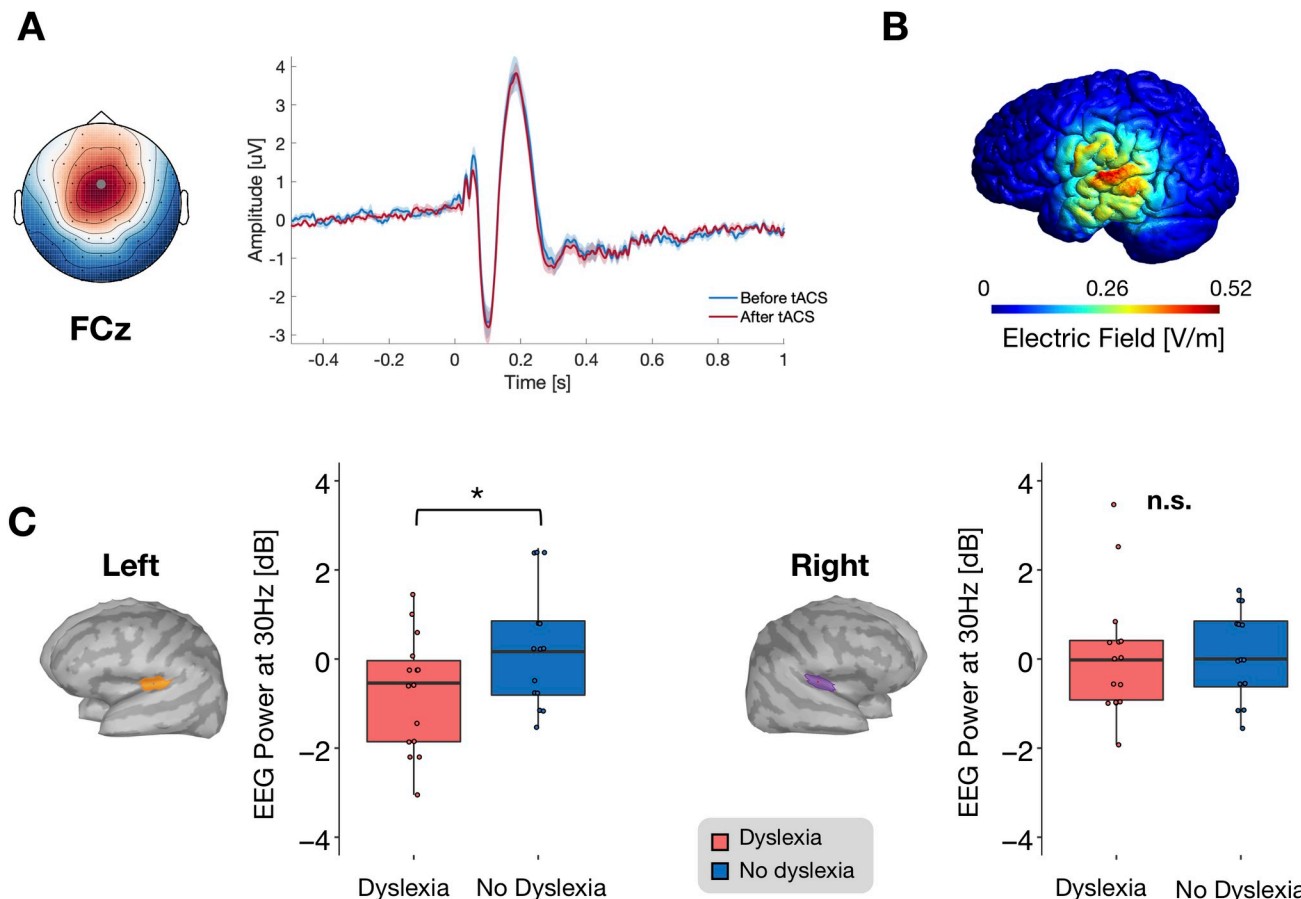

**Fig 1. Auditory-evoked response at the level of scalp electrodes, tACS-induced electric field reconstruction, and group difference at baseline at 30 Hz.** (A) The average response over time (dyslexia group, all AM frequencies considered) showed the typical auditory N100 and P200 components and related topography. The strongest peak-to-peak response amplitude was recorded at electrode FCz for both the N100 and P200 components. (B) Simulation of the electric field induced by tACS using a high-definition 4 × 1 electrode configuration, displayed on a head model from an individual subject. The model has been obtained with the freeware software SimNIBs [37] and Gmsh (www.gmsh.info). The electric field elicited by the tACS can be observed selectively in the left hemisphere, more prominently on auditory brain regions, whereas in the contralateral hemisphere, the resultant electric field can be estimated to zero (see S8 Fig). (C) Source reconstruction of the 30-Hz EEG response to 30-Hz AM before tACS. In the left auditory cortex, the dyslexia group showed a reduced response as compared to the no-dyslexia group (left); no difference between the groups was found in the right hemisphere (right). Numerical data used to generate this figure can be found at https://osf.io/6j49q/. Significance is denoted with * for $p < 0.05$. AM, amplitude-modulated; EEG, electroencephalography; tACS, transcranial alternating current stimulation.

performance following tACS across stimulation conditions (sham, 30 Hz, 60 Hz) and between the 2 groups (dyslexia, no dyslexia), we considered the between-time–measurements difference, i.e., before and after tACS stimulation. We ran a repeated-measures ANOVA with group (dyslexia, no dyslexia) as a between-subjects factor and stimulation condition (sham, 30 Hz, 60 Hz) as a within-subject factor. We first assessed the effect of tACS on phonemic awareness, considering the number of errors in the repeated phonemes at the pseudoword test. We found a main effect of group ($F_{1,28} = 11.4$, $p < 0.01$, $\eta^2_p = 0.28$) and stimulation conditions ($F_{2,56} = 3.16$, $p = 0.05$, $\eta^2_p = 0.1$) and a significant interaction between group and stimulation conditions ($F_{2,56} = 3.96$, $p < 0.05$, $\eta^2_p = 0.12$, Fig 3A). Post hoc tests revealed that the improvement after 30-Hz tACS was stronger in the dyslexia than in the control group ($T_{74.8} = -4.27$, $p < 0.001$, d = 1.68, False Discovery Rate [FDR]), as well as for sham ($T_{56} = -2.62$, $p < 0.05$, d = 0.73, FDR) and 60 Hz tACS ($T_{56} = -3.64$, $p < 0.01$, d = 0.75, FDR) in the dyslexia group.

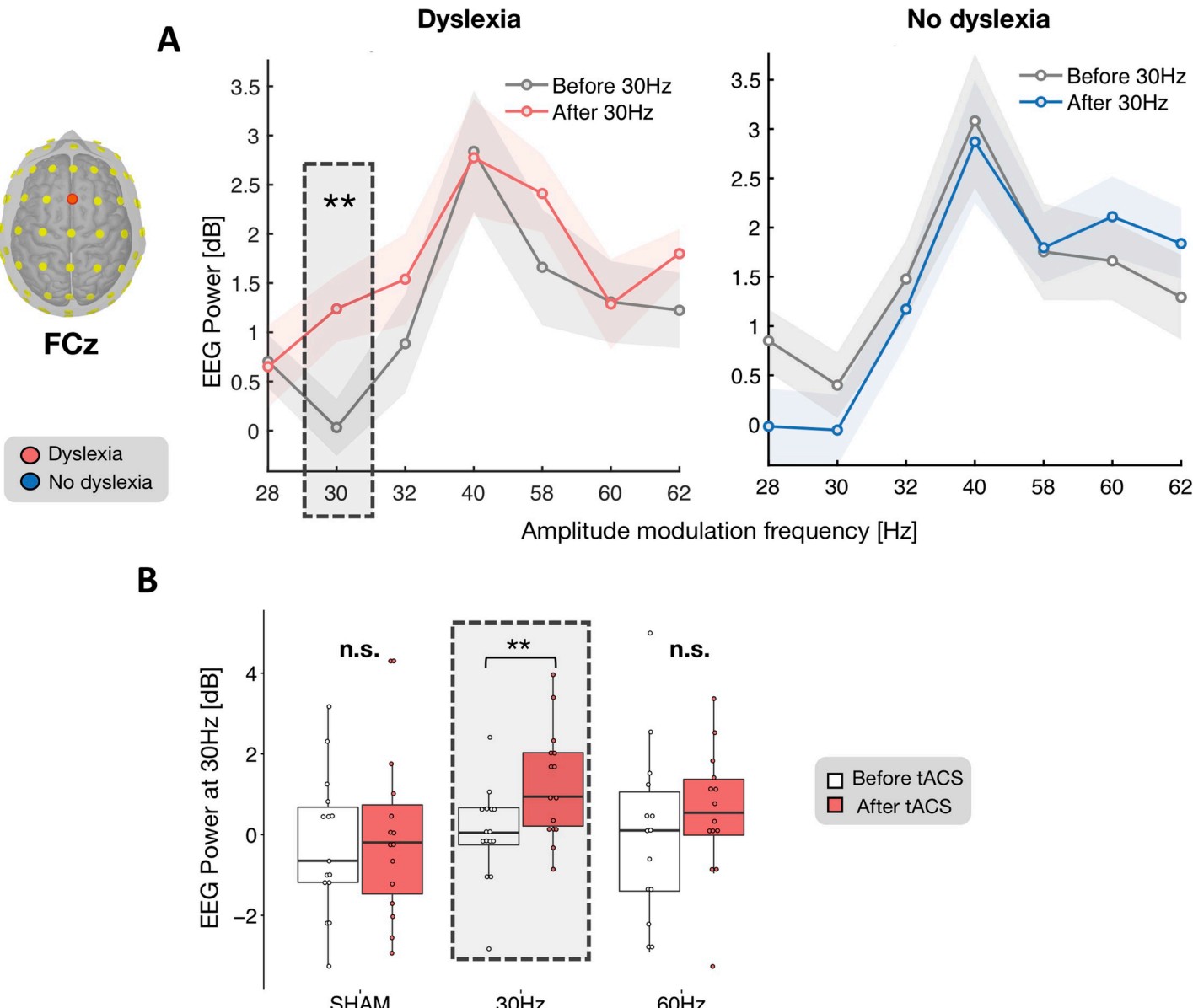

**Fig 2. tACS-induced power modulation in ASSR at the scalp level.** ASSR in power (dB) to pure sounds modulated in amplitude (AM) with specific frequencies (from 28 Hz to 62 Hz, x-axis) recorded at electrode FCz. This electrode was chosen because it displayed the strongest evoked response in the time domain (Fig 1A). We considered the EEG power at the frequency corresponding to that of the AM pure tones (e.g., 30-Hz EEG power in response to 30-Hz AM tones). Both the dyslexia (A, left) and no-dyslexia groups (A, right) showed the strongest response for 40-Hz AM tones and the weakest for 30-Hz AM tones. In the dyslexia group, the 30-Hz tACS elicited a selective 30-Hz power increase for the 30-Hz AM sounds (A). This effect was absent in the sham and 60-Hz conditions in the dyslexia (B) as well as in the no-dyslexia (A, right) group. Numerical data used to generate this figure can be found at https://osf.io/6j49q/. Significance is denoted with ** for $p < 0.01$. AM, amplitude-modulated; ASSR, auditory steady-state response; EEG, electroencephalography; tACS, transcranial alternating current stimulation.

Although several models agree on a phonological deficit in dyslexia, it is still debated whether the underlying core impairment arises at the phonemic or syllabic level [26]. To address the specificity of the phonemic improvement after 30-Hz stimulation, we also evaluated syllable short-term memory from the pseudoword test. We compared the number of omitted or wrongly repeated syllables (absolute values) as measured before and after the 30-Hz tACS and found that tACS did not improve syllable processing ($T_{14} = 0.43$, $p > 0.05$, d = 0.11;

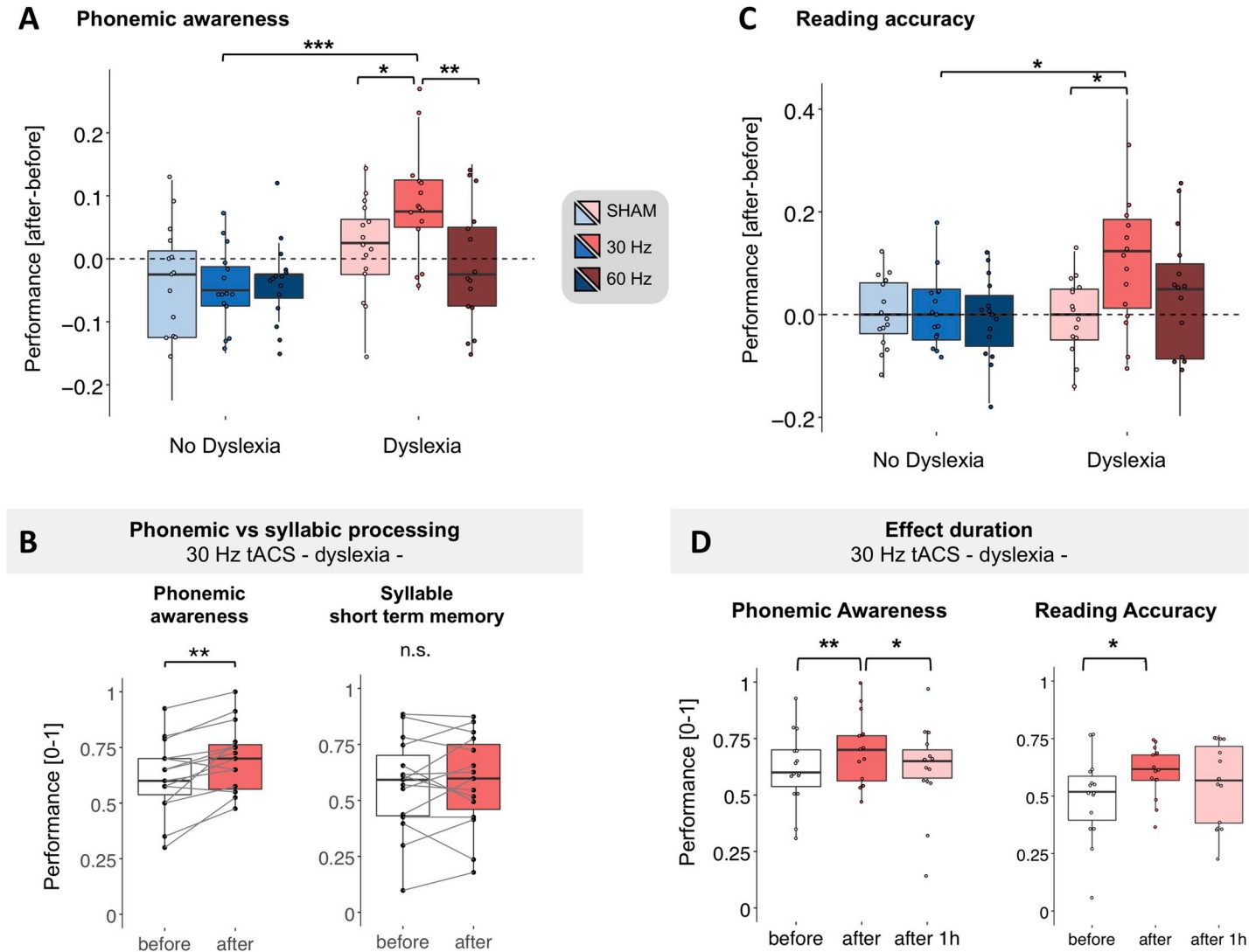

**Fig 3. tACS stimulation effectiveness: Behavioral results.** Changes in performance occurring after tACS on phonemic awareness (A) and reading accuracy (C) for each tACS condition (sham, 30 Hz, 60 Hz) in the no-dyslexia (shades of blue) and dyslexia (shades of red) groups. Each variable is obtained as the difference between the before versus after tACS measurements. For each of the 2 metrics, we performed a repeated-measures ANOVA with group (dyslexia, no dyslexia) as a between-subjects factor and stimulation condition (sham, 30 Hz, 60 Hz) as a within-subject factor. For both variables, the improvement after 30-Hz tACS was stronger in the dyslexia than in the control group, as well as after the sham in the same group and 60 Hz for the phonemic awareness index only. The improvement in phonemic awareness immediately after 30-Hz tACS in the dyslexic group was not accompanied by changes in syllable short-term memory (B). In the dyslexia group, performance increased after 30-Hz tACS for both phonemic awareness and reading accuracy and decreased 1 hour after for the phonemic awareness index only (D). Numerical data used to generate this figure can be found at https://osf.io/6j49q/. Significance is denoted with * for $p < 0.05$, ** for $p < 0.01$, *** for $p < 0.001$. tACS, transcranial alternating current stimulation.

Fig 3B). Note, however, that since we only probed a tACS effect at 30 Hz, this result does not contradict the hypothesis that slower brain rhythms could also play a role in dyslexia by interfering at the syllabic level [38,39].

To address whether phonemic awareness improvement following 30-Hz tACS stimulation in the dyslexia group could ultimately result in or be associated with better reading performance, we extracted and analyzed separately changes in reading accuracy (see the "Language assessment" section in the Methods) and speed from the text-reading task. With respect to poststimulation reading accuracy (Fig 3C), we found a main effect of group ($F_{1,28} = 4.89$,

$p < 0.05$, $\eta^2_p = 0.09$) and of the stimulation condition ($F_{2,56} = 3.13$, $p = 0.051$, $\eta^2_p = 0.1$), as well as a marginal group × stimulation interaction ($F_{2,56} = 2.5$, $p = 0.09$, $\eta^2_p = 0.08$). Given this trend and the results found for phonemic awareness, we tested for the same pattern in reading accuracy. As for phonemic awareness, the reading accuracy improvement after 30-Hz tACS was stronger in the dyslexia than the control group ($T_{83.6} = -3$, $p < 0.05$, d = 0.96) and stronger than in the sham condition in the dyslexia group ($T_{56} = -3.2$, $p < 0.05$, d = 0.67). A similar trend was observed for the 60-Hz tACS condition ($T_{56} = -2.37$, $p = 0.06$, d = 0.45).

To further characterize the behavioral improvement with respect to the initial individual reading level of the participants, we correlated the gain in phonemic awareness and in reading accuracy induced by 30-Hz tACS with the ECLA16+ (Évaluation de Compétences de Lecture chez l'Adulte de plus de 16 ans) score [40]. With respect to phonemic awareness, we found a markedly negative relationship across groups (r = −0.48, $p < 0.01$; S3A Fig), although within-groups correlations were not significant (dyslexia: r = 0.09, $p = 0.73$; no dyslexia: r = 0.36, $p = 0.18$). This result indicates that the intervention was more effective in boosting phonological performance in participants who initially had poor reading skills (lower ECLA16+) than in good readers. For the latter, the effect of tACS on phonological processing was even slightly disruptive. The gain in reading accuracy after 30-Hz tACS also correlated with the ECLA16+ score across groups (reading accuracy: r = −0.50, $p < 0.01$, S3B Fig) but not within groups (dyslexia: r = −0.33, $p = 0.23$; no dyslexia: r = 0.03, $p = 0.9$), confirming a stronger effect in the poorer readers.

Although we did not find any significant change in performance in reading speed, a qualitative analysis of the results (S4A Fig) suggests that reading speed was slowed down after all stimulation conditions regardless of dyslexia severity. These results could indicate that the phonemic awareness improvement translated into reading accuracy, whereas reading speed was altered by a nonspecific tACS effect.

Last, we found no effect of tACS in global phonological performance, as assessed by the spoonerism test (S4B Fig).

Given the specific effectiveness of 30-Hz tACS in enhancing phonemic awareness and reading accuracy in the dyslexia group, we then investigated the persistence of the effects by considering the data acquired 1 hour after the stimulation. We applied the same statistical approach as for the previous data set (i.e., difference after/before): we ran a repeated-measures ANOVA with tACS condition (sham, 30 Hz, 60 Hz) as a within-subjects factor and group (dyslexia, no dyslexia) as a between-subjects factor, considering as a dependent variable the performance difference between the 1-hour–after and before conditions. For both measures, results did not reveal any statistically significant results for the tACS condition, group, or interaction between the 2 factors (S5 Fig).

The decline of the immediate tACS effect was confirmed by a one-way repeated-measures ANOVA with time (before, after, 1 hour after) as a within-subjects factor on the data (absolute values) of the dyslexia group for the 30-Hz tACS condition for the phonemic awareness and reading accuracy index. Both analyses revealed a main effect of time (phonemic awareness: $F_{2,28} = 7.51$, $p < 0.01$, $\eta^2_p = 0.35$, reading accuracy: $F_{2,28} = 4.35$, $p < 0.05$, $\eta^2_p = 0.23$, Fig 3D). Performance increased after tACS for both indexes (phonemic awareness: $T_{28} = 3.78$, $p < 0.01$, d = 0.97; reading accuracy: $T_{28} = 3.08$, $p < 0.05$, d = 0.79, FDR corrected) and decreased 1 hour after for the phonemic awareness index only ($T_{28} = -2.56$, $p < 0.05$, d = 0.66, FDR corrected).

Having established that subjects with dyslexia could benefit from 30-Hz tACS, we examined the neural activity underpinning the behavioral improvement occurring selectively at this frequency of stimulation.

To take advantage of a maximal S/N (signal-to-noise ratio), we first used electrode FCz, the sensor displaying the strongest evoked response (Fig 1A), to assess tACS effects on the 30-Hz

auditory response separately in the 2 groups. tACS at 30 Hz elicited a significant power increase in the 30-Hz ASSR in the dyslexia group ($T_{14}$ = 3.7, $p < 0.01$, d = 0.97; Fig 2A left, S6 Fig), whereas no effect was found in controls ($T_{13}$ = 0.8, $p$ = 0.43, d = 0.21; Fig 2A right). Critically, we found no enhancement of ASSR power in the dyslexia group after 60 Hz ($T_{13}$ = 0.991, $p > 0.05$, d = 0.256; Fig 2B) or sham ($T_{14}$ = 0.458, $p > 0.05$, d = 0.118; Fig 2B) tACS. Overall, these results indicate that 30-Hz tACS was selectively effective at the frequency that was disrupted in the group with dyslexia. The absence of such an effect in controls shows that 30-Hz tACS was ineffective when oscillatory activity was already present.

We further evaluated the 30-Hz tACS effect on the corresponding 30-Hz ASSRs in the dyslexia group separately in each of the 2 abovementioned ROIs (i.e., auditory cortex and STG) by performing a $2 \times 2$ repeated-measures ANOVA with time (before/after) and hemisphere (left/right) as factors. In the auditory cortex, we found a significant power increase after 30-Hz tACS (main effect of time, $F_{1,14}$ = 13.71, $p < 0.01$, $\eta^2_p$ = 0.49; Fig 4A). We then tested whether the 30-Hz tACS applied over the left auditory cortex had a local impact on the neural activity in this region. We found a significant increase in oscillatory response in the left auditory cortex ($T_{14}$ = 3.4, $p < 0.05$, d = 0.88, FDR corrected; Fig 4A). The effect was not present in the right auditory cortex ($T_{14}$ = 1.6, $p > 0.05$, d = 0.42, FDR corrected; Fig 4A), even though there was no time × hemisphere interaction.

Finally, we probed whether the effect propagated to surrounding regions in subjects with dyslexia. We performed the same analysis in the STG and found a significant time × hemisphere interaction ($F_{1,14}$ = 10.6, $p < 0.01$, $\eta^2_p$ = 0.43; Fig 4B). Like in the auditory cortex, 30-Hz tACS increased the response in the left hemisphere ($T_{14}$ = 2.8, $p < 0.05$, d = 0.72, FDR corrected). Furthermore, left tACS tended to reduce the 30-Hz response in the right STG ($T_{14}$ = −1.5, p = 0.1, d = 0.4, FDR corrected) to the extent that after 30-Hz tACS, ASSRs were significantly stronger in the left than the right STG ($T_{14}$ = −3.1, $< 0.05$, d = 0.8, FDR corrected), reversing the pattern we observed before stimulation ($T_{14}$ = 1.8, $p$ = 0.12, d = 0.45, FDR corrected). These results indicate that tACS did not only boost 30-Hz neural activity in left auditory regions but also reinstated its left hemispheric dominance in subjects with dyslexia.

In the control group, the same analyses performed in both ROIs did not reveal a significant difference between hemisphere and time. We then sought for a relationship between the severity of dyslexia (ECLA16+) and the gain in the auditory cortex and found a negative correlation in the left hemisphere (r = −0.46, $p$ = 0.013; Fig 4C, left), but not the right (r = −0.14, $p$ = 0.44). We also observed that the phonemic awareness gain was markedly related to the 30-Hz ASSR power increase in the left auditory cortex (r = 0.36, p = 0.06; Fig 4C, right).

## Discussion

We show that tACS delivered at 30 Hz could enhance phonological abilities in individuals with dyslexia. This finding has important implications in basic aspects of speech-processing research because the relation between low-gamma neural activity and phonemic encoding has so far only been either conjectured [18,41,42] or inferred using correlational studies [22,43,44]. Selectively reinstating low-gamma auditory activity in subjects with dyslexia, in whom it is disrupted, offers a unique opportunity to address this important issue. By showing that 30-Hz, but not 60-Hz, tACS boosted phonemic, but not syllabic, perception, we can now confidently argue for a causal link between the presence of low-gamma activity in left auditory cortex and basic phoneme encoding via temporal integration windows of about 35 ms [18].

This results evidently also constitutes an interesting promise for potentially normalizing phonological processing in subjects with dyslexia. Previous attempts at improving reading

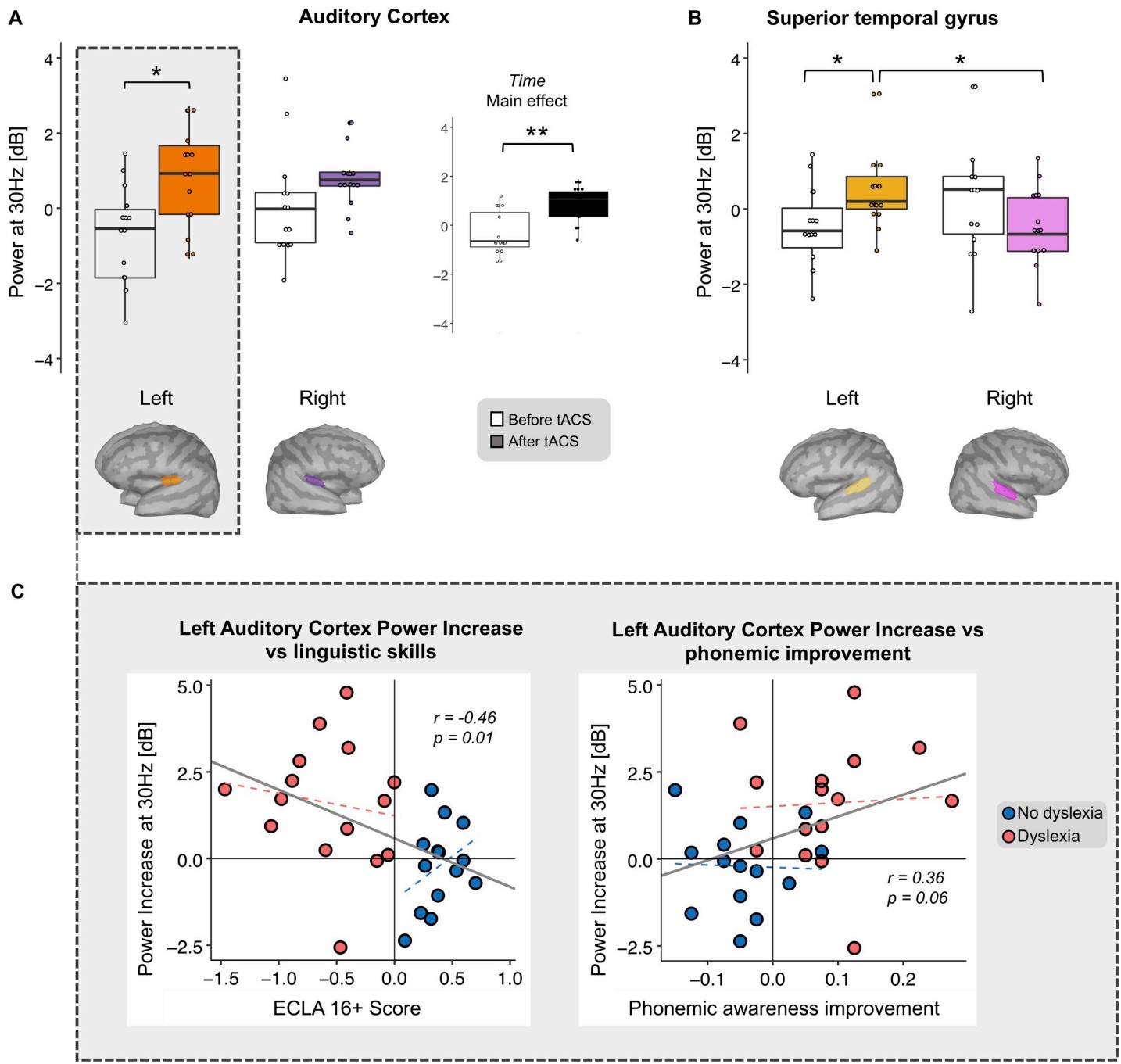

**Fig 4. A 30-Hz EEG power increase after 30-Hz tACS in auditory cortex and STG.** Average power (dB) of ASSRs to 30-Hz AM pure tones over 2 ROIs in each hemisphere, before (white whisker plots) and after (colored whisker plots) 30-Hz tACS. Differences were tested by considering as fixed effects time (before/after tACS) and hemisphere (left/right) separately in the auditory cortex (A) and the STG (B). In auditory cortex, 30-Hz tACS increased responses bilaterally (A, right: average activity of both hemispheres for each time, main effect of time), an effect driven by a significant power increase in left auditory cortex (A, left). A significant interaction between time and hemisphere in the STG (B) revealed that EEG power after tACS was significantly higher than before on the left hemisphere. In addition, after the stimulation, power in the left hemisphere was stronger than in the right one. The relation between 30-Hz power gain in left auditory cortex and behavioral variables was considered across groups (solid gray lines). Changes in power correlated negatively with dyslexia severity (C, left: r = −0.46, p = 0.01) and positively (trend) with phonemic improvement (C, right: r = 0.36, p = 0.06). Within-group trend lines are displayed with dashed lines (red: dyslexia group, r = −0.15, p = 0.59 for ECLA, r = 0.05, p = 0.85 for phonemic awareness; blue: control group, r = 0.35, p = 0.22 for ECLA, r = −0.036, p = 0.9 for phonemic awareness). Numerical data used to generate this figure can be found at https://osf.io/6j49q/. Significance is denoted with * for p < 0.05, ** for p < 0.01. AM, amplitude-modulated; ASSR, auditory steady-state response; ECLA16+, Évaluation de Compétences de Lecture chez l'Adulte de plus de 16 ans; EEG, electroencephalography; ROI, region of interest; STG, superior temporal gyrus; tACS, transcranial alternating current stimulation.

efficiency and phonological processing using neuromodulation already gave encouraging results [6–8]. However, because these studies have either used transcranial direct current stimulation (tDCS), which does not target specific brain rhythms, or tACS but without showing a specific effect on oscillatory activity, the exact neural bases of performance enhancement could not be specified. The current results show that phonological processing improved via a focal and specific enhancement of 30-Hz oscillatory activity in left auditory cortex. This finding confirms previous ones in other disorders, showing that tACS can indeed influence neuronal activity in a frequency- and location-specific manner [45,46]. Phonological performance improvement hence likely reflects changes in neuronal entrainment towards the frequency of the tACS stimulation [45,47].

Interestingly, the facilitating tACS effect on phonological processing was most pronounced in individuals who had poor reading skills, whereas a slightly disruptive effect was observed in very good readers. This result could be related to previous observations that very fast readers may have developed neural reading strategies that largely skip phonological processing [48]. In these subjects, a 30-Hz tACS might interfere negatively with reading by weighing too much on phonological processing.

The observation that tACS was selectively effective in individuals with dyslexia confirms previously shown performance-dependent effects of transcranial electrical stimulation, in which a larger behavioral improvement occurred more strongly in poorly performing than in proficient individuals [49–51]. As observed experimentally [52] and predicted by computational models [31,53], tACS efficacy depends on the gain in endogenous oscillation power. Accordingly, the current data associate a stronger phonological improvement to a higher oscillation power gain in left auditory cortex. This is consistent with previous observations that oscillatory power increases after tACS only when baseline oscillatory activity is low [52,54], which might reflect a ceiling effect whereby high/normal oscillation power cannot be further enhanced (at least with the well-tolerated current intensities). Such a ceiling effect in controls could provide a plausible explanation for a stronger positive 30-Hz tACS effect in poorer readers.

From a neurobiological perspective, tACS performance-dependent effectiveness might reflect individual changes in the excitatory/inhibitory balance, a variable that has previously been associated with reading performance [55]. The presence of variable levels of dysregulation of cortical excitability in dyslexia could explain why the same stimulation did not lead to comparable effects in all subjects.

Reinstating 30-Hz activity had an immediate impact on the phonological deficit, as it improved both pseudoword reading and text-reading accuracy by about 15%. However, a single 20-minute tACS exposure was not sufficient to induce a long lasting low-gamma power enhancement. Short-lived effects (<1 hour) following acute tACS intervention are extensively reported in the literature [56–58]. They are agued to reflect short-term plasticity (STP), which declines over a period of half an hour [59,60]. STP selectively amplifies synaptic signals in a frequency-dependent manner, presumably inducing spike-timing–dependent plasticity via N-methyl-D-aspartate (NMDA) receptors [59–61]. However, whether the beneficial tACS effects on phonological processing globally reflects a normalization of glutamatergic activity, previously shown to be excessive in poor readers [55], remains to be demonstrated.

STP per se is not sufficient to induce long-term potentiation (LTP) lasting from hours to days [59,60]. Longer-lasting changes can be obtained by repeating the tACS simulation over several consecutive days [62]. With tDCS, cumulative effects can be observed after only 3 daily 20-minute exposures [63] and can last up to 1 month after the first session. A 6-month effect has been observed after 6 days of tACS combined with targeted behavioral training [64]. We could hence expect to induce a longer-lasting improvement of phonemic processing and to

stabilize tACS effects through consecutive, repetitive stimulation sessions, and these effects could even be potentiated by combining tACS administration with phonological training [65].

Although a single intervention acutely improved performance, a chronic treatment could be expected to not only durably normalize the phonemic sampling capacity but also induce large-scale modifications of the reading network [12]. Our results already suggest that larger-scale effects could be induced by tACS because it induced a relateralization of 30-Hz responses to the left temporal cortex. A similar immediate interhemispheric effect was previously reported in the context of interactions between primary motor cortices using other neuromodulation techniques [66,67]. If stabilized by repeated stimulation, such a large-scale effect might also confer on the 30-Hz tACS procedure the interesting advantage of (re)circumscribing phonemic processing to the left hemisphere, an important functional property for improving reading speed in the longer run. Future protocols might also exploit dose-dependent neural entrainment by tACS to tailor stimulation intensity to individual neural deficits in gamma activity [46].

An important finding of the current study is that tACS improved reading accuracy (see S2 Table for individual number of reading errors), but not reading speed. Distinct effects on reading speed and accuracy likely reflect that the acute 20-minute stimulation acted primarily on phonemic encoding locally in bilateral temporal cortices, but not (yet) on phonological downstream processing. The access to phonemic representation via interactions with the left inferior frontal cortex remained presumably unchanged by the stimulation. Such critical interactions might, however, improve if the phonemic representation format were durably normalized by repeated tACS; if this is the case, an improvement of reading speed could be expected, in accordance with the phonological deficit hypothesis.

In summary, the current results demonstrate for the first time, to our knowledge, the causal role of low-gamma oscillatory activity in phonemic processing and further show the selective impact of targeted tACS on the phonological deficit in dyslexia. This new line of research offers interesting perspectives for promoting plasticity in the reading network via the correction of basic properties of the auditory cortex [68].

## Methods

### Participants

Fifteen adults previously diagnosed with dyslexia (dyslexia group: 13 women, 2 men, mean age 27.4 years, SD ± 9, range 18–47) and 15 fluent readers (no-dyslexia group: 11 women, 4 men, mean age 25.6 years, SD ± 7.8, range 18–47) took part in our study (S3 Table and S7 Fig).

The experimental paradigm was approved by the local Ethics Committee (Commission cantonale d'éthique de la recherche, project #15–264) and was performed in accordance with the Declaration of Helsinki. The study has been registered retrospectively in a publicly accessible clinical trials registry approved by WHO on www.clinicaltrials.gov (NCT04277351). All participants provided written informed consent and received a financial compensation for their participation.

The exclusion criteria were a history of brain injury, neurological or psychiatric disorders, and the presence of invasive electronic implants. All participants had normal hearing acuity as assessed with an audiogram (pure-tone frequency threshold between 250 and 8,000 Hz) and adequate nonverbal intelligence (standard score above 80 on Raven's progressive matrices [69]). The 2 groups did not present any differences in nonverbal intelligence ($T_{28} = 1.37$, $p > 0.05$, d = 0.5).

To be included in the dyslexia group, participants were required to present a history of dyslexia previously assessed by a speech-language therapist that had to be confirmed by the ECLA16+ test during the inclusion day.

## Experimental paradigm

The protocol was spread over 4 experimental days (S1 Fig): an inclusion day to assess language and cognitive performance and 3 experimental days, during which tACS was administered. The tests lasted approximately 5–6 h/day, amounting to a total duration of 22 hours of experimental time per participant (S1 Fig).

The auditory stimuli used in the language tests and during the EEG experiment were delivered binaurally using insert earphones (ER-2; Etymotic Research, Elk Grove Village, IL, USA) at 70–75 dB SPL via a graphical user interface developed in MATLAB (version 2015; The MathWorks, Natick, MA, USA). Responses to the language tests were recorded with a microphone and subsequently analyzed by a certified linguist.

**Inclusion day (day 1).**   The dyslexia diagnosis was confirmed during the inclusion day using the ECLA16+ test, a standardized tool to evaluate reading proficiency (positive values indicates high performance [40]). The test includes multiple subtests assessing phonological awareness, short-term memory, and reading skills. Further statistical analysis (2-tailed unpaired $t$ test between groups) confirmed that the dyslexia group performed worse than fluent readers for all individual skills tested (phonological awareness: $T_{28} = 7.6$, $p < 0.0001$, d = 2.78; verbal short-term memory: $T_{28} = 4.2$, $p < 0.001$, d = 1.55; reading skills: $T_{28} = 4.9$, $p < 0.0001$, d = 1.79; S3 Table). Subjects additionally performed a rapid automatized naming (RAN) test, which confirmed that the dyslexia group had reduced lexical access (2-tailed unpaired $t$ test between groups, $T_{28} = 5.62$, $p < 0.0001$, d = 2.06; S3 Table). Last, for familiarization purposes, participants underwent a training session with the same battery of custom-designed language tests as those used during the experimental days (see below).

**Experimental days (days 2–4).**   During each of the following experimental days (day 2 to 4), participants received 1 of 3 tACS stimulation conditions (30 Hz, 60 Hz, and a sham condition in which subjects received no stimulation) in a counterbalanced order across subjects. Experimental days were separated by at least 10 days. Each experimental day included measurements at 3 time points—before, after, and 1 hour after tACS, respectively—in order to reveal both immediate and potentially longer-lasting tACS effects. At the beginning of each of the 3 time measurements, we recorded ASSRs to pure tones modulated in amplitude at various frequencies (28–62 Hz) by means of EEG, followed by language tests.

## Language assessment

Language tests were specifically designed for this study by a certified linguist in order to probe those skills that are most strongly impaired in dyslexia, namely phonological awareness, short-term memory, and reading speed and accuracy. These skills were assessed using 3 tests: pseudoword repetition, spoonerism, and text reading (S1 Table). The development of a custom-designed solution was necessary to ensure the diversity of linguistic material across experimental time measurements (before, after, 1 hour after) and minimize learning effects that might occur over repeated measurements within the same experimental day. All versions of each of the 3 subgroups of tests (1 for each measurement, before, after, and 1 hour after) had similar phonological and syntactic features, as well as the same lexical frequency.

To assess equal difficulty in all 3 tests and exclude learning effects within 1 day, we ran a pilot study in 28 fluent readers, different from those included in the main experiment. Participants performed the 3 subgroups of language tests in the same order and with the same timing as in the experimental paradigm. Because statistical tests did not show a significant difference across measurements, we assumed that there was no difference in terms of difficulty within the test material and no detectable learning effect over 3 consecutive measurements (one-way repeated-measures ANOVA). This triple battery of tests was then used during each of the 3

experimental days (1 for each tACS condition), leading to a total of 3 repetitions per subtest. To account for potential learning effects occurring over the 3 experimental days, we regressed out the slope relative to the improvement as described in the section "Statistical analysis of language tests."

**Pseudoword repetition.**   This test assesses the participant's phoneme representation (i.e., the ability to retrieve and recall each phoneme) and syllabic short-term memory (i.e., the ability to recall syllable sequences as accurately as possible [70,71]). A total of 30 different items were included in the test; these were different within each subtest and between the 3 time measurements (i.e., before, after, and 1 hour after tACS), leading to a total of 90 items. Each pseudoword consisted of 4 to 8 syllables. Participants were instructed to repeat single pseudowords immediately after hearing them. We evaluated phonemic awareness by considering errors in phoneme repetition due to a perception deficit, i.e., regarding a single articulatory feature (voice onset timing or place of articulation). Syllable short-term memory was computed by taking into account the number of non- or wrongly repeated syllables.

**Spoonerism.**   This task assesses phonological and lexical structure awareness [40,72]. Participants were presented with 2 regular words and were asked to repeat them after transposing the first phoneme (e.g., hand–pig becomes pand–hig). From this test, we calculated an accuracy index as the total number of correctly inverted words and a speed score as the average time required to perform one phoneme inversion. A global phonological awareness score was computed by averaging the 2 abovementioned indexes.

**Text reading.**   This test evaluates reading skills, taking into account lexical knowledge (written word identification) and decoding abilities (lexical orthography to phonology conversion) [73]. Participants were asked to read as fluently as possible a scientific text for 3 minutes [74]. Reading speed corresponded to the number of read words and reading accuracy to the number of phoneme-to-grapheme errors.

## EEG recording and stimuli

Along with the language tests, we recorded EEG using a 64-channel setup (Brain Products GmbH, Gilching, Germany) before, immediately after, and 1 hour after each tACS condition (30 Hz, 60 Hz, and sham). Because the EEG cap was not removed between time measurements, a passive-electrode EEG system was chosen to ensure stable impedances throughout the entire experimental day (approximately 6 h). Electrode AFz was used as ground contact and FCz as reference. Raw signals were sampled at 1 kHz using proprietary software (Recorder, BrainProducts GmbH).

During the EEG recordings, we presented AM sounds to entrain brain oscillations in a frequency-specific manner (frequency-tagging) and measured the resultant ASSR in the steady period beginning 500 ms after sound onset and following the initial auditory-evoked potential. Frequency-tagging probes the capacity of auditory cortical regions to synchronize to an auditory stimulus with constant amplitude [75,76]. Pure tones (carrier frequency: 1,000 Hz) modulated in amplitude at specific frequencies (28, 30, 32, 40, 58, 60, and 62 Hz) were presented for 1.5 s. Because of technical issues, the auditory stimuli at 40 Hz were not presented in 1 participant of the no-dyslexia group during the sham session. For each AM condition, the stimulus presentation was repeated 40 times with a 3.5-s interstimulus interval. Each EEG-ASSR block lasted approximately 25 minutes, during which participants remained seated in front of a screen placed 1 meter away from their forehead, displaying a muted video of their choice. In order to minimize artifacts on the EEG traces, participants were asked to avoid eye and body movements.

## tACS

tACS (Starstim, Neuroelectrics, Barcelona, Spain) was delivered via 5 electrodes (Pistim, Neuroelectrics; $\pi cm^2$ contact area) placed over the left auditory cortex and integrated into the 64-channel EEG cap to ensure invariant position between participants. Electroconductive gel was used to ensure optimal conductance between the electrodes and the scalp. Stimulation electrodes were arranged in a $4 \times 1$ ring configuration at TTP7h, FTT9h, FCC5h, CPP5h, and TPP9h, with the central one delivering an alternating current below 2 mA and the surrounding 4 electrodes delivering one-fourth of the central electrode's current in the opposite polarity (S1 Fig). The $4 \times 1$ electrode configuration is a well-established experimental procedure that delivers the electrical stimulation focally to a specific brain region [77].

The tACS intensity was tuned separately for each participant, starting from 0.6 mA and increasing by steps of 200 μA until perception threshold, with a maximum peak-to-peak intensity of 2 mA. The current was then reduced below that threshold, and this value was kept constant throughout the entire duration of the experiment, with 20 s of ramp-up and ramp-down. The average peak-to-peak stimulation intensity across participants was 1.1 mA for the 30-Hz tACS and 1.2 mA for the 60-Hz condition. In the sham condition, 30-Hz tACS was delivered only during the ramp-up and ramp-down periods (20 s); no current was delivered during the 20-minute intervention. The tACS stimulation lasted 20 minutes, a duration that is common in cognitive neuroscience research [32]. During the tACS, as for the EEG recording, participants sat comfortably in front of a screen, watching a muted video of their choice but without receiving any auditory stimulation.

Overall, participants' reports indicate that they could not tell whether they received real or sham tACS: out of 30 participants, only 7 correctly suspected not having received a stimulation during the sham condition, whereas 18 participants did not suspect any sham tACS and believed they had received electrical stimulation over the 3 experimental days.

At the end of each stimulation session, subjects were debriefed about side effects (pain, tingling or any skin sensation, warming, fatigue, or attentional difficulties), associated with tACS by rating the level of discomfort on a scale between 0 and 10 (S7 Fig). A repeated-measures ANOVA with group (dyslexia/no dyslexia) as a between-subjects factor and tACS condition (sham/30 Hz/60 Hz) as a within-subjects factor revealed a main effect of stimulation condition ($F_{2,28} = 9.1$, $p < 0.001$, $\eta^2_p = 0.23$). Post hoc analyses showed weaker reported side effects for the sham than the 30-Hz ($T_{29} = 3$, $p < 0.01$, $d = 0.54$) and 60-Hz conditions ($T_{29} = 4.3$, $p < 0.001$, $d = 0.78$). We found no difference in the reported negative perceptions between the 2 active stimulation groups. Given the absence of significant difference in the reported side effects between the 30-Hz and 60-Hz conditions and between groups, we can exclude that the enhanced behavioral performance following the 30-Hz tACS, selective in the dyslexia group, could have been influenced by discomfort sensations.

## Data analysis

**Statistical analysis of language tests.**   All behavioral data were rescaled between 0 (low performance) and 1 (high performance) by taking into consideration the entire pool of 30 subjects separately for each subtest. Even though no learning effect over the 3 time measurements (i.e., before, after, and 1 hour after) within the same day was observed in the pilot behavioral study run in 28 independent subjects, we identified a trend for improvement over experimental days. For this reason, we considered, for each participant and each test separately, the performance during the first session (before stimulation) of each experimental day. We computed a slope value reflecting the theoretical improvement that might occur between the first and last experimental day and regressed it out from the data set (see S4 Table for a quantitative evaluation of the learning across experimental days).

In order to investigate the impact of tACS on the different components contributing to reading skills, we analyzed phonemic awareness (pseudoword reading), reading accuracy and speed, and global phonological awareness (spoonerism test) separately. We considered the difference in performance observed immediately after versus before and 1 hour after versus before. Separately for these 2 time points (immediately after and 1 hour after), we ran a mixed repeated-measures ANOVA with group (dyslexia, no dyslexia) as a between-subjects factor and tACS condition (sham, 30 Hz, 60 Hz) as a within-subjects factor. First, we used this approach to investigate immediate effects (after/before) in all metrics (phonemic awareness, reading accuracy and speed, global phonological awareness) and then to restrict the analysis on the 1-hour–after change in those metrics that showed a modulation immediately after the tACS.

We also probed whether potential changes in phonemic awareness were accompanied by variations in syllable processing and analyzed syllable short-term memory for the specific tACS condition and time measurement of interest (30 Hz, immediately after tACS).

Following up on the mixed ANOVA, post hoc comparisons were performed based on estimated marginal means and corrected for multiple comparisons using the FDR method. In accordance with our hypothesis, we restricted between-groups comparisons to performance difference of the same tACS condition.

To investigate whether potential tACS-induced behavioral changes were related to dyslexia severity, we performed Pearson correlation analyses between the ECLA16+ score and the performance difference at the time point of interest.

All behavioral analyses were performed with R (2018).

**EEG preprocessing and analysis.** EEG data preprocessing was conducted using the EEGlab v14.1.2 [78] and SASICA [79] toolboxes within the MATLAB (The MathWorks) environment. Signals were down-sampled at 500 Hz and filtered using a Hamming windowed-sinc FIR filter between 1 and 70 Hz. EEG epochs were defined from 1 s prestimulus to 2 s poststimulus. Epochs contaminated by strong muscular artifacts were excluded by visual inspection. Noisy channels were automatically identified using a custom-written MATLAB script based on the presence of high-frequency activity, inspected visually, and subsequently removed. When a single channel exhibited an epoch-specific artifact, the signal was interpolated for that epoch only. Data were then re-referenced to average reference. Independent component analysis (ICA) was applied to the epoched data set, the dimensionality of which was previously reduced by principal component analysis to 32 components. Artifactual components were identified using a semiautomatic method based on, among others, measures of focal channel topography, autocorrelation, and generic discontinuity available through the SASICA toolbox. After ICA-based denoising, the artifact-contaminated channels initially removed were interpolated using spherical splines, and all epochs were visually inspected again and rejected if artifacts remained. Data from 1 participant in the no-dyslexia group and data relative to the 60-Hz session in 1 participant in the dyslexia group were excluded from the analysis because of a strong artifact contamination. EEG data acquired 1 hour after the tACS conditions had poor signal quality because of the duration of the acquisition (approximately 4 hours after the EEG setup). In a few cases, we even had to interrupt the EEG recording session. For this reason, the 1-hour–after EEG data sets have not been taken into consideration.

Subsequent data analyses were performed using Fieldtrip [80] and Brainstorm [81] toolboxes, together with custom scripts.

**Surface EEG space analysis.** First, we computed the grand average signal over time for the dyslexia group to identify the auditory-evoked potential and the scalp electrode exhibiting the strongest peak-to-peak amplitude. We restricted the surface space analysis to this electrode, FCz. The time–frequency transform in both surface and source spaces was estimated using a

discrete Fourier transform (Hanning taper; 1–70 Hz; 1 Hz steps). At the surface level, we quantified the ASSR for each AM condition as the average power at the AM frequency between 500 ms and 1,500 ms after sound onset, normalized with respect to the 1 s prestimulus baseline. For each AM condition, we tested for differences between the measurements obtained before and immediately after the stimulation (2-tailed paired $t$ test).

**Source space analysis.** The distributed source space, consisting of a 15,000-vertex mesh of the cortical surface, was obtained from the segmentation of a template MRI (Colin27, MNI). Using the OpenMEEG [82] implementation in Brainstorm software [81], we generated a 3-layer EEG Boundary Element Method model consisting of the inner skull, outer skull, and the scalp surfaces, with corresponding conductivity values of 0.33:0.0125:0.33 S/m, respectively. Within this model, the source activity of dipoles distributed over the cortical surface was estimated by a minimum-norm approach with noise normalization (dSPM) and constrained orientation (depth weighting order: 0.5, maximal amount: 10; S/N: 3; noise covariance regularization: 0.1). Spatial smoothing (FWHM 3mm) was applied for displaying the average source activity (S2 Fig). We outlined 2 ROIs of 79–96 vertices in the primary auditory cortex and the STG for each hemisphere. The average time–frequency spectrum over each ROI was calculated from source activity with the same method as for the surface analysis. To assess whether we could replicate previous findings [22], we tested for differences between the 2 groups at baseline in the ASSR at 30 Hz by performing a repeated-measures ANOVA with group (dyslexia/no dyslexia) as within-subjects factor and hemisphere (left/right) as between-subjects factor. We then tested our a priori hypothesis that activity is reduced in the left auditory cortex in the dyslexia group relative to the control group by comparing the 2 groups separately in the left and right primary auditory cortex (2-tailed unpaired $t$ test). Subsequent source space analyses were performed to address putative tACS effects between measurements obtained before and after the 30-Hz tACS. Differences in the laterality of power response were investigated separately for each of the 2 ROIs with a $2 \times 2$ ANOVA, using time (before/after) and hemisphere (left/right) as factors. We investigated the relationship between changes in power following the 30-Hz tACS and behavioral measurements such as the performance improvement in phonemic awareness after the 30-Hz tACS and the dyslexia severity (ECLA16 + score) by performing Pearson correlation analyses.

## Supporting information

**S1 Fig. Experimental paradigm.** Each participant (15 with dyslexia, 15 without dyslexia) undertook 4 days of testing. During the inclusion day (day 1), the severity of dyslexia was assessed with the ECLA16+ test together with evaluating baseline performance on 3 language tests. These included the pseudoword repetition test, spoonerism test, and text reading, probing, among other things, errors at the phonemic and syllabic level. This allowed us to evaluate phonological awareness, verbal short-term memory, and reading accuracy, as well as the ability to convert lexical orthography to phonology. During each of the following experimental days (days 2–4), one of the 3 tACS stimulation conditions (sham, 30 Hz, and 60 Hz) was administered, with the tACS condition order counterbalanced across subjects. The stimulation lasted 20 minutes and was delivered by means of 5 tACS electrodes organized as a $4 \times 1$ ring and centered over the left auditory cortex. Within each experimental day, performance on the 3 language tests was evaluated at 3 time points: before, immediately after, and 1 hour after tACS stimulation. At each time point and for each tACS condition, EEG was recorded by means of a 64-channel cap. We used ASSRs to AM pure-tone sounds with fixed frequencies (from 28 Hz to 60 Hz) to entrain brain oscillations in a frequency-specific manner. AM, amplitude-modulated; ASSR, auditory steady-state response; ECLA16+, Évaluation de Compétences de Lecture

chez l'Adulte de plus de 16 ans; EEG, electroencephalography; tACS, transcranial alternating current stimulation.
(TIF)

**S2 Fig. Auditory-evoked response in the source space.** Source localization revealed that the neural generators of the evoked response at short latencies (approximately 60 ms poststimulus onset) are located over the primary auditory cortex bilaterally.
(TIF)

**S3 Fig. Relationship between language skills and behavioral improvement.** Changes in performance occurring after 30-Hz tACS (difference after/before) for both phonemic awareness (A, r = −0.48, $p$ = 0.007) and reading accuracy (B, r = −0.5, $p$ = 0.005) show a strong negative relationship with language skills (ECLA16+, higher values indicate better reading skills) across groups (solid gray lines). Within-group trend lines are displayed with dashed lines (red: dyslexia group, r = 0.09, $p$ = 0.73 for phonemic awareness, r = −0.32, $p$ = 0.23 for reading accuracy; blue: control group, r = 0.36, $p$ = 0.18 for phonemic awareness, r = 0.034, $p$ = 0.9 for reading accuracy). Numerical data used to generate this figure can be found at https://osf.io/6j49q/. ECLA16+, Évaluation de Compétences de Lecture chez l'Adulte de plus de 16 ans; tACS, transcranial alternating current stimulation.
(TIF)

**S4 Fig. Immediate tACS stimulation effects on reading speed and global phonological awareness.** Changes in performance immediately after tACS for each condition (sham, 30 Hz, 60 Hz) in the no-dyslexia (shades of blue) and dyslexia (shades of red) groups are calculated as the difference with respect to the performance measured before the tACS. For reading speed and global phonological awareness, we performed a repeated-measures ANOVA with group (dyslexia, no dyslexia) as a between-subjects factor and stimulation condition (sham, 30 Hz, 60 Hz) as a within-subject factor and found no statistically significant changes. Reading speed: group, $F_{1,28}$ = 0.02, $p$ > 0.05, $\eta^2_p$ = 0.001; tACS condition, $F_{2,56}$ = 0.38, $p$ > 0.05, $\eta^2_p$ = 0.013; interaction group × tACS condition, $F_{2,56}$ = 2.38, $p$ > 0.05, $\eta^2_p$ = 0.07. Global phonological awareness: group, $F_{1,28}$ = 1.2, $p$ > 0.05, $\eta^2_p$ = 0.05; tACS condition, $F_{2,56}$ = 1.57, $p$ > 0.05, $\eta^2_p$ = 0.05; interaction group × tACS condition, $F_{2,56}$ = 1.09, p > 0.05, $\eta^2_p$ = 0.03. Numerical data used to generate this figure can be found at https://osf.io/6j49q/. tACS, transcranial alternating current stimulation.
(TIF)

**S5 Fig. tACS stimulation effects on phonemic awareness and reading accuracy 1 hour after stimulation.** Changes in performance occurring 1 hour after tACS on phonemic awareness (A) and reading accuracy (B) for each tACS condition (sham, 30 Hz, 60 Hz) in the no-dyslexia (shades of blue) and dyslexia (shades of red). Each parameter is calculated as the difference with respect to the performance measured before tACS. For each of the two metrics, we performed a repeated-measures ANOVA with group (dyslexia, no dyslexia) as a between-subjects factor and stimulation condition (sham, 30 Hz, 60 Hz) as a within-subject factor and found no statistically significant changes. Phonemic awareness: group, $F_{1,28}$ = 0.02, $p$ > 0.05, $\eta^2_p$ = 0.001; tACS condition, $F_{2,56}$ = 0.08, $p$ > 0.05, $\eta^2_p$ = 0.003; interaction group × tACS condition, $F_{2,56}$ = 1.63, $p$ > 0.05, $\eta^2_p$ = 0.05. Reading accuracy: group, $F_{1,28}$ = 2.93, $p$ > 0.05, $\eta^2_p$ = 0.11; tACS condition, $F_{2,56}$ = 2.07, $p$ > 0.05, $\eta^2_p$ = 0.09; interaction group × tACS condition, $F_{2,56}$ = 1.23, $p$ > 0.05, $\eta^2_p$ = 0.04. Numerical data used to generate this figure can be found at https://osf.io/6j49q/. tACS, transcranial alternating current stimulation.
(TIF)

**S6 Fig. Difference at the topographical level in power response to auditory stimuli at 30 Hz measured before versus immediately after the 30-Hz tACS in the dyslexia group.** The topography displays $t$ values for each of the 64 channels obtained through 2-tailed paired-samples $t$ test. Positive values indicate a stronger response after the 30-Hz tACS; black dots highlight electrodes with $p < 0.05$. Numerical data used to generate this figure can be found at https://osf.io/6j49q/. tACS, transcranial alternating current stimulation.
(TIF)

**S7 Fig. Demographic information and reported side effects of tACS.** Individual demographic information, language skills measured during the inclusion day (ECLA16+), nonverbal intelligence (Raven matrix), and reported side effects of tACS for each stimulation condition (sham/30 Hz/60 Hz). This included physical sensations—such as pain, warming, tingling—and anxiety, fatigue, and attention. The reported side effects were stronger for the active conditions as compared with the sham stimulation. There was no difference between the dyslexia and no-dyslexia groups. Numerical data used to generate this figure can be found at https://osf.io/6j49q/. Significance is denoted with ** for $p < 0.01$, *** for $p < 0.001$. ECLA16+, Évaluation de Compétences de Lecture chez l'Adulte de plus de 16 ans; tACS, transcranial alternating current stimulation.
(TIF)

**S8 Fig. Simulated electric field induced by the 4 × 1 high-definition tACS applied to the left hemisphere (see Fig 1), displayed on an individual head model.** Results show null electric field in the right hemisphere, contralateral to the stimulation site. The coronal view at the peak strength shows focal activity over left auditory regions. tACS, transcranial alternating current stimulation.
(TIF)

**S1 Table. Battery of language tests.** Three language tests were designed by a certified linguist to probe phonological processing, short-term memory, and reading skills (speed and accuracy). The pseudoword repetition test consisted of repeating nonlexical words that contain existing syllables in French. The spoonerism test consisted in transposing the first phoneme of 2 words, chosen to have similar phonological and syntactic features and with the same lexical frequency across the 3 subtests, one for each time measurement. Text reading consisted of reading a scientific text for 3 minutes.
(TIF)

**S2 Table. Dyslexia diagnosis during the inclusion day.** Scores in the dyslexia and no-dyslexia groups for the ECLA16+ diagnosis test and the RAN test evaluating phonological awareness, reading skills, and short-term memory. Negative values represent performance below average, positive values performance above average; all values are z-scores. Stars indicates significant differences between the dyslexia and no-dyslexia groups. Significance is denoted with *** for $p < 0.001$. ECLA16+, Évaluation de Compétences de Lecture chez l'Adulte de plus de 16 ans; RAN, rapid automatized naming.
(TIF)

**S3 Table. Individual number of reading errors in the dyslexia group at each time measurement during the 30-Hz tACS condition.** Each row corresponds to one participant in the dyslexia group. tACS, transcranial alternating current stimulation.
(TIF)

**S4 Table. Regression of potential learning effect across the 3 experimental days.** Individual slopes were calculated by considering the performance measured "before" stimulation

separately for each variable of interest (phoneme, syllables, etc.). The table displays the average performance during the first experimental day (asterisks indicates number of errors instead of absolute performance), the average slopes, and learning rate across participants. The slopes were significantly different from zero (one-sample $t$ test) for all variables except for the Phonemic Awareness index $T_{29} = 0.57$, $p > 0.05$, d = 0.1). They were similar in the 2 groups (dyslexia, no dyslexia) in all variables but phonological awareness response speed (spoonerism test: $T_{28} = 2.39$, $p < 0.05$, d = 0.87, 2-sample unpaired $t$ test). Please note that this index was not analyzed individually but pooled together with the spoonerism accuracy in order to compute a Global Phonological Awareness index.
(TIF)

## Acknowledgments

We thank P. Mégevand for helpful discussion about the data analysis and comments on the manuscript; S. Martin for comments on the manuscript; and C. Pacoret, F. Hummel, and the Neurostimulation platform of the Biotech Campus for technical advice. We also thank E. Pool for helpful discussion on the statistical approach.

## Author Contributions

**Conceptualization:** Anne-Lise Giraud.

**Data curation:** Silvia Marchesotti, Johanna Nicolle.

**Formal analysis:** Silvia Marchesotti, Johanna Nicolle, Isabelle Merlet, Luc H. Arnal, Anne-Lise Giraud.

**Funding acquisition:** Anne-Lise Giraud.

**Investigation:** Silvia Marchesotti, Johanna Nicolle.

**Methodology:** Silvia Marchesotti, Johanna Nicolle, Isabelle Merlet, Anne-Lise Giraud.

**Software:** Silvia Marchesotti, Johanna Nicolle, Isabelle Merlet.

**Supervision:** Anne-Lise Giraud.

**Visualization:** Silvia Marchesotti.

**Writing – original draft:** Silvia Marchesotti, Johanna Nicolle, John P. Donoghue, Anne-Lise Giraud.

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
