## [Editor Report · Decision Letter 0]

18 Feb 2020

Dear Dr Marchesotti, 

Thank you for submitting your manuscript entitled "Selective enhancement of low-gamma activity by tACS improves phonemic processing and reading accuracy in dyslexia" for consideration as a Research Article by PLOS Biology.

Your manuscript has now been evaluated by the PLOS Biology editorial staff as well as by an Academic Editor with relevant expertise and I am writing to let you know that we would like to send your submission out for external peer review.

However, before we can send your manuscript to reviewers, we need you to complete your submission by providing the metadata that is required for full assessment. *Please also take this opportunity to upload your updated manuscript file that mentions the trial registration.*

To this end, please login to Editorial Manager where you will find the paper in the 'Submissions Needing Revisions' folder on your homepage. Please click 'Revise Submission' from the Action Links and complete all additional questions in the submission questionnaire.

Please re-submit your manuscript within two working days, i.e. by Feb 20 2020 11:59PM.

Kind regards,

Hashi Wijayatilake, PhD

Managing Editor

on behalf of

Gabriel Gasque, Ph.D.,

Senior Editor

PLOS Biology

---

## [Decision Letter · Decision Letter 1]

31 Mar 2020

Dear Dr Marchesotti,

Thank you very much for submitting your manuscript "Selective enhancement of low-gamma activity by tACS improves phonemic processing and reading accuracy in dyslexia" for consideration as a Research Article at PLOS Biology. Your manuscript has been evaluated by the PLOS Biology editors, by an Academic Editor with relevant expertise, and by three independent reviewers. You will note that reviewers 1 and 3, Matthew Krause and Benedikt Zoefel, respectively, have signed their comments. 

In light of the reviews (below), we will not be able to accept the current version of the manuscript, but we would welcome re-submission of a much-revised version that takes into account the reviewers' comments. We cannot make any decision about publication until we have seen the revised manuscript and your response to the reviewers' comments. Your revised manuscript is also likely to be sent for further evaluation by the reviewers.

We expect to receive your revised manuscript within 2 months. 

**IMPORTANT - SUBMITTING YOUR REVISION**

Your revisions should address the specific points made by each reviewer. As you will see, all reviewers agree your study is potentially significant and relevant. However, reviewers 1 and 2 raise serious overlapping concerns regarding the statistical analyses. Having discussed these comments with the Academic Editor, we think that for further consideration, these concerns should be thoroughly and satisfactorily addressed and your original conclusions must remain.

Please submit the following files along with your revised manuscript:

*Re-submission Checklist*

*Published Peer Review*

*PLOS Data Policy*

*Blot and Gel Data Policy*

Sincerely,

Gabriel Gasque, Ph.D., 

Senior Editor

PLOS Biology

REVIEWS:

Reviewer #1, Matthew R. Krause: Thank you for inviting me to review "Selective enhancement of low-gamma activity by tACS improves phonemic processing and reading accuracy in dyslexia", by Marchesotti, Nicolle, and colleagues (PBIOLOGY-D-20-00231R1). In this manuscript, the authors report that applying 30 Hz tACS to left auditory cortex improves phonemic awareness and reading accuracy in dyslexic subjects immediately following stimulation. This is consistent with a theory of oscillatory auditory processing and lays the groundwork for a possible intervention for a difficult condition with few other treatment options. I therefore expect that these results will be of interest to the diverse readership of PLoS Biology. 

 In particular, the design and rationale of this brain stimulation experiment is one of the best that I have read. The authors have a clear theoretical reason for choosing 30 Hz tACS plus some experimental support showing a low gamma deficit in the left auditory cortex of dyslexic patients, which they replicate in Figure 1. Possible stimulation confounds are controlled for by blinding and comparing against both sham stimulation and another active stimulus (60 Hz) at the same site. Stimulation has the predicted effect on neural activity, alleviating the 30 Hz EEG power deficit and improving behavioral performance on reading and phonological tasks, but not syllable-based ones. This is a solid careful design for a neurostimulation experiment. My current research is focused on brain stimulation rather than dyslexia, but the linguistic tasks and interpretation also seem appropriate based on what I know and the cited references.

 That said, I do have a few concerns about the analytic approach. The first is that several analyses use two t-tests or ANOVAs to separately test for an effect (e.g., changes in phonemic awareness before/after tACS) separately in each group (e.g., dyslexic/non-dyslexic). Inferences are then drawn based on which groups did or did not show significant differences. This approach is generally not advisable (see Gelman and Stern, 2006). Instead, the authors should directly test the changes of interest (e.g., does the change in reading accuracy differ after sham/30/60 Hz). This could also be tackled with a mixed-effects model with interactions. I suspect that the theoretically-interesting comparisons will still pan out, but it would be better be careful. 

 Secondly, Figure 2C/2E and 4C show a single correlation coefficient/line, which is presumably calculated from all of the subjects and meant to show differential improvements in the most severely affected subjects. I think it would be better to add separate lines and coefficients for each group (e.g., by adding red and blue lines/numbers). I would like to rule out an issue where the correlation coefficients just reflect differences in the group means rather than individual subjects. Specifically, the healthy controls will tend to be clustered towards the right (since ECLA 16+ measures reading proficiency) and below the x-axis (since they are already near ceiling per Figure 2); dyslexic patients will be towards the left, but less downward as the ceiling effect is smaller. In a quick simulation, I found that this might be enough to produce some spurious correlation even if there is no effect. This could be ruled out by showing that the correlations hold up within each group (or equivalently, the right patterns of main effect + interactions). 

 The mechanistic discussion in Lines 241-258 is perhaps a little speculative, given what we currently (don't) know about tACS aftereffects. While I agree that some form of plasticity is likely involved, I'm not sure the paper really requires a deep dive into possible molecular mechanisms until they are better understood. On the other hand, Line 259 may undersell their results—an intervention that only helps during or immediately after stimulation could potentially still be a big win for patients with dyslexia. 

 Overall, I think this is a well-designed experiment and only needs minor changes to the analysis, described above. A few more suggestions/trivial errors are listed below.

Other comments:

Line 131 (and others): I found it confusing that the time point factor (before/immediately after/1 hr) was called "session", which could also refer to sham/30/60 Hz stimulation, as it does in the legend of Figure S1. If you do not want to rename the factor, I suggest explicitly defining it around Line 128: "During each day, subjects' language abilities were tested in three sessions: before stimulation, immediately afterwards, and one hour later." 

Line 165-6: "reading speed was slowed-down immediately after all stimulation conditions". The data in Figure S3 suggests things are a little more complicated. Specifically, dyslexic subjects show slowing after both stimulation (but not sham) conditions; non-dyslexics showed slowing after sham and 60 Hz. Some of this is likely due to confounds (subjects are more tired/distractable, etc right after stimulation). This possibility should be tested more directly by comparing the changes in speed across conditions, instead of drawing inferences between which before/after comparisons are/are not significant. 

Line 173: "20 min tACS [is] not sufficient to positively impact speech production". Is there a reason to think it would? If not, I would suggest rephrasing it.

Line 389, 392-3: tACS systems vary in how they report stimulation intensity (peak-to-peak, 0 to max, RMS). I believe the system used here reports 0-max, so it might be clearer to report the stimulation intensity as "starting from ±0.8 mA" or "with a maximum intensity of 4 mA (peak to peak)." 

Line 385: The PISTIM electrodes' inner diameter is 2 cm (so that the total area is πcm²) rather than 1.2 cm reported here (which is the outer radius, but includes the 0.2 mm inactive plastic lip).

* Line 416: Rescaling between 0 and 1 is probably the right thing to do for the data analysis, but it makes it hard to gauge the practical effect of this intervention. It might be worth slipping an unscaled version of the reading effect size (like 'subjects misread 10 fewer words') into the results or discussion. 

Line 441: Pearson, rather than Person, correlation. 

Figure 1 (c. Line 733). Is the brain shown in Panel B one of the participants or an atlas? Either is fine, but it should be indicated in the caption. 

Figure 2B (c. Line 747). It is unclear which 'after' condition is meant by the light gray bars. The text hints at 'immediately after', but the color scheme in 1A/1D suggests '1 hr after'.

Figure 2,4, S2, S (c. Lines 747, 780, 819, 834). It is not clear what the small numbers in Panel 2C, 2D, 4C, S2C, S3B, and S3C indicate. 

Figure 4 (c. Line 780) It would be nice if the Y axes in Panel C matched (they are close, but not quite the same).

Reference:

Gelman, A., & Stern, H. (2006). The difference between "significant" and "not significant" is not itself statistically significant. The American Statistician, 60(4), 328-331.

Reviewer #2: In the study "Selective enhancement of low-gamma activity by tACS improves phonemic processing and reading accuracy in dyslexia" Marchesotti and colleagues investigated the effect of 20 min 30 Hz tACS, as well as 60 Hz tACS and sham tACS, over the left auditory cortex on phonological processing and reading accuracy in 15 adult dyslexic and 15 healthy control participants. 

In result the authors confirm reduced 30 Hz ASSR responses in the left auditory cortex for dyslexic participants compared to healthy controls. Importantly they further show that 30 Hz tACS improved phonemic awareness as well as reading accuracy in the dyslexic but not in the control group from pre to post measurement. Furthermore, tACS induced improvements in phonemic awareness and reading accuracy were related to the severity of the dyslexic symptoms across all participants. Finally, 30 Hz tACS increased the 30 Hz ASSR response in dyslexic but not in healthy control participants. 

This is a highly relevant study addressing a very interesting and timely topic. On top of this, the manuscript is theoretically well embedded and proportional and concisely written. 

However, unfortunately the manuscript in its present form comprises several methodological issues that prevent a publication in PLOS Biology. In sum, the statistical approach is invalid, data smoothing/handling is insufficient explained and the process under investigation remains elusive. Please find below a more detailed list of concerns/problems/questions that might help improving the manuscript.

Statistics:

In its present form the statistic approach looks like "cherry-picking" in it selects only a subset of available data and statistical tests that support their claims. In their experimental design the authors included 2 (groups) by 3 (tACS conditions [30Hz, 60Hz, sham]) by 3 (measurement [pre, after, 1h after]) factors but none of the performed analyses does include all of them. In contrast the authors performed 2 (group) x 3 (measurement) ANOVAs separately for each stimulation condition for behavioral data 1 (phonemic awareness) BUT 3 (pre, after, 1h after) x 3 (30Hz, 60Hz, sham) ANOVAs separately for each group for behavioral data 2 -4 (accuracy, speed text reading, spoonerism) without valid justification. 

The correlations performed include all participants but the results are interpreted for the dyslexic patients only. Is there a valid correlation in patients only or does the correlation mainly represent the group differences? 

Analogously, when "confirming" the 30 Hz response deficit (ASSR) in the left AC a 2 (group) x 2 (hemisphere) ANOVA should have been performed resulting in a significant interaction (not done). Even though I agree that in this case there is a strong and valid a-priori hypothesis allowing for direct testing. But why are the pre-to-post effects of 30 Hz tACS on the 30 Hz ASSR tested separately for each group and each stimulation condition? Finally, while the authors eventually performed a 2 (pre, post) x 2 (hemisphere)) ANOVA for the ROI analysis (only for 30 Hz tACS, separately for each group) they continue to invalidly post-hoc test pre-to-post changes in the left and right AC without a significant interaction of both parameters. 

Finally, EEG data for the "1h-after" measure are completely ignored. 

Data processing:

Do the authors really assume that the results from a separate pilot group performing the behavioral test-battery 3 times are comparable to the data of the experimental groups performing the behavioral test-battery NINE times? 

The information given for the performed correction of the observed repetition effects needs to be improved. How strong was the repetition effect and how was this "corrected"?

Does any statistical result survive if the test would have been performed without reducing the variance of the behavioral data between subjects? Why wasn't this correction applied to the ECLA scores as well? 

How many different items have been presented during each of the 9 tests for the pseudo-word repetition task? Since the task produced the main behavioral effect of the present study, this information seems essential. The manuscript indicates that the same 30 items were presented each time? If this is true, there needs to be a large repetition effect, especially since several participants showed an - admittedly non-verbal - IQ of >120. 

Further:

The EEG positions seem to belong to a 128 channel EEG recording cap and might be referenced accordingly. 

There is no "anode" or "cathode" during tACS. 

The information about the frequency of stimulation (30Hz or 60 Hz, or separately for both) for the threshold measurement is missing. What was the mean stimulation intensity?

The stimulation regime was not blinded neither for the experimenter nor for the participants in the end.

Conceptually: 

1. Which process is intended to be manipulated by tACS? Is the 30Hz ASSR deficit related to a PERCEPTUAL impairment?

2. There seems to be also inconsistency with respect to the hypothesis of altered gamma oscillations in dyslexia. While the authors initially mention a dysfunctional (to high) sampling rate as the cause of dyslexia they eventually modulate the amplitude of low-gamma oscillations. 

3. How should an improvement in phonemic awareness instantaneously result in an improved reading performance?

Reviewer #3, Benedikt Zoefel: The present manuscript examines whether transcranial alternating current stimulation (tACS) at 30 Hz improves phonological processing in dyslexic individuals. The notion underlying the study is the previous demonstration that (1) dyslexic adults show a deficit in oscillatory activity at a frequency corresponding to the phonetic rate in human speech (~30 Hz) and that (2) tACS can boost oscillatory activity. 

Indeed, the authors report that (1) dyslexic individuals show a reduced ("auditory steady-state") response (ASSR) to amplitude-modulated sounds at 30 Hz (as compared to healthy individuals); (2) this response is selectively boosted (i.e. only at 30 Hz) after 30-Hz tACS; (3) the enhancement in 30-Hz ASSR goes along with improvements in phonological processing; (4) this effect is not present after sham stimulation or stimulation at 60 Hz.

I can see no major flaw in this manuscript. The manuscript is structured very clearly and logically. Experimental design and analyses are sound. Results are exciting given that they reveal potential treatment for dyslexic individuals in the future. I do have a few comments which I hope can be used to improve the manuscript even further.

It is unclear what participants are instructed to do during the stimulation. Did they attend the electric stimulation passively, without sensory stimulation? The current applied during tCS modulates membrane potentials only by a very small amount (~0.3 mV) and therefore needs to interact with ongoing neural activity to produce a measurable outcome (Stagg & Nitsche, 2011). If auditory cortices are inactive during stimulation (i.e. ongoing neural activity is reduced), I wonder how the authors can explain the efficacy of the stimulation - could they comment on this, please.

The authors frame their observed effect as a demonstration of tACS-induced manipulation of oscillatory activity at 30 Hz. However, they simply show that 30-Hz tACS modulates EEG responses to amplitude-modulated sounds at 30 Hz (i.e. 30-Hz ASSR). There is no demonstration that genuine oscillatory activity is modulated (e.g., it is not shown that resting-state oscillations at 30 Hz are altered, nor that these are reduced in dyslexic individuals). It is possible that tACS merely modulates neurons that respond preferentially to a stimulus frequency corresponding to that of the applied tACS. I therefore recommend to tune down the claim that oscillations are involved; I do not think it needs to be very prominent anyway (if tACS can be used to boost phonological processing, it is only of secondary importance if this entails a manipulation of neural oscillations).

A paper is cited entitled "Transcranial electrical stimulation improves phoneme processing in developmental dyslexia". I wonder whether this paper needs to be given more attention, given the similarity of the research question. Could the authors please elaborate on the difference between their study and the cited one.

The authors show 30-Hz ASSR aftereffects only for electrode FCz; It would be interesting to see the effect for the whole topography, as tACS might affect neural activity measured at electrodes other than that where we see the strongest ASSR.

---

## [Decision Letter · Decision Letter 2]

7 Jul 2020

Dear Dr Marchesotti,

Thank you for submitting your revised Research Article entitled "Selective enhancement of low-gamma activity by tACS improves phonemic processing and reading accuracy in dyslexia" for publication in PLOS Biology. I have now obtained advice from the original reviewers and have discussed their comments with the Academic Editor. You will note that reviewer 3, Benedikt Zoefel, has signed his comments. 

Based on the reviews, we will probably accept this manuscript for publication, assuming that you will modify the manuscript to address the remaining points raised by reviewer 1, which we have discussed with the Academic Editor and think should be implemented. Please also make sure to address the data and other policy-related requests noted at the end of this email.

We expect to receive your revised manuscript within two weeks. 

Your revisions should address the specific points made by each reviewer. Please include in your submission:

In addition to the remaining revisions and before we will be able to formally accept your manuscript and consider it "in press", we also need to ensure that your article conforms to our guidelines. A member of our team will be in touch shortly with a set of requests. As we can't proceed until these requirements are met, your swift response will help prevent delays to publication.

*Copyediting*

*Published Peer Review History*

*Early Version*

*Submitting Your Revision*

Sincerely,

Gabriel Gasque, Ph.D., 

Senior Editor

PLOS Biology

DATA POLICY:

Note that we do not require all raw data. Rather, we ask for all individual quantitative observations that underlie the data summarized in the figures and results of your paper. For an example see here: http://www.plosbiology.org/article/info%3Adoi%2F10.1371%2Fjournal.pbio.1001908#s5

These data can be made available in one of the following forms:

Regardless of the method selected, please ensure that you provide the individual numerical values that underlie the summary data displayed in the following figures:

Figures 1AC, 2A-D, 3AB, 4A-C, S3AB, S4AB, S5AB, S6, and S7. 

Please also ensure that each figure legend in your manuscript include information on where the underlying data can be found and ensure your supplemental data file/s has a legend.

Reviewer remarks:

Reviewer #1: Thank you for inviting me to re-review "Selective enhancement of low-gamma activity by tACS improves phonemic processing and reading accuracy in dyslexia" by Marchesotti, Nicolle, and colleagues (PBIOLOGY-D-20-00231R2). As in my initial review, I found the paper to be an interesting and well-written manuscript. The Reviewers' comments and edits have largely addressed the questions I raised in the last round, and the manuscript now seems ready for publication in PLoS Biology.

I have a few final suggestions, but all of them are completely optional. 

In the interests of transparency, I would prefer that both the pooled data and separate dyslexic/control correlations be reported in Figure 4c (maybe with smaller, dashed lines). It's up the reviewers. 

The caption on Figure 1B could be revised to indicate that it's a single individual's brain. It might also be useful to include the contralateral hemisphere (or say in the text that no fields above X were detected), as effects on both hemispheres are discussed in the paper.

In the responses to Reviewers 1 and 2, the 1hr time point is described as a control. Different papers report a wide range of tACS after-effects, from nearly zero to few surprising reports suggesting that a single session produces changes detectable 3-4 hours later, so I am a little skeptical that it is a valuable control. However, if it's what the authors intend, that point could be made more strongly in the paper—I read it more as an interesting follow-up question.

Reviewer #2: This was a thorough revision and the authors have been very responsive to my feedback. They present interesting data from a sound and scientific approach that is a valuable contribution. In conclusion, I recommend its publication.

Reviewer #3, Benedikt Zoefel: I thank the authors for addressing all of my comments. I think they have produced a strong manuscript.

---

## [Editor Report · Decision Letter 3]

4 Aug 2020

Dear Dr Marchesotti,

On behalf of my colleagues and the Academic Editor, Simon Hanslmayr, I am pleased to inform you that we will be delighted to publish your Research Article in PLOS Biology. 

Early Version

PRESS 

Kind regards,

Vita Usova

Publication Editor, 

PLOS Biology

on behalf of

Gabriel Gasque,

Senior Editor

PLOS Biology